# Association between shortened maternal and fetal telomere length and abnormal fetal development

Océane Coudrieu[1], Zangbéwendé Guy Ouedraogo[1,2], Denis Gallot[3], Amélie Delabaere[3], Lauren Veronese[1,4], Eleonore Eymard-Pierre[1], Andrei Tchirkov[1,4], Carole Goumy [1,5]*

1 Cytogénétique Médicale, CHU Estaing; Université Clermont Auvergne, UFR de Médecine et professions paramédicales, Clermont-Ferrand, France, 2 CNRS, Inserm, iGReD, Université Clermont Auvergne, Clermont-Ferrand, France, 3 Unité de Médecine Fœtale, CHU Clermont-Ferrand, CHU Estaing, Clermont-Ferrand, France, 4 EA7453 CHELTER, Clonal Heterogeneity, Leukemic environment, Therapy resistance of chronic leukemias, Université Clermont Auvergne, Clermont-Ferrand, France, 5 INSERM U1240 Imagerie Moléculaire et Stratégies Théranostiques, Université Clermont Auvergne, Clermont-Ferrand, France

* cgoumy@chu-clermontferrand.fr

## Abstract

A number of intrinsic, maternal and environmental factors have been linked to the risk of fetal developmental anomalies. In a previous study, we showed that telomere length (TL) was notably reduced in amniotic fluid when the fetus exhibited a developmental anomaly. In this new study, we measured the fetal and maternal TL for 75 evolutive pregnancies with congenital malformation. We also measured the TL of 50 pregnant women without fetal anomalies and 50 non-pregnant control women who had at least one child with normal development. In fetal samples, telomeres were significantly shortened in cases with congenital anomalies compared to controls (n = 93) (P < 0.0001). Interestingly, age-adjusted maternal TL was also significantly reduced in these cases (P < 0.01). Receiver operating characteristic (ROC) analysis showed that maternal TL, at the optimal cut-off value, identified cases of congenital anomalies with 92% specificity and 73% sensitivity. In addition, fetal and maternal TL were correlated, with 15% to 38% of the variance in fetal TL attributable to maternal TL. Telomere shortening can lead to increased sensitivity to various maternal exposure factors and may contribute to compromised organogenesis, possibly due to inadequate cell proliferation or genomic instability. Measuring maternal TL during the periconceptional period could serve as a useful predictive biomarker for assessing the risk of fetal developmental anomalies.

## Introduction

About 2–4% of pregnancies feature isolated or multiple congenital anomalies. The European Surveillance of Congenital Anomalies (EUROCAT) estimates that the prevalence of major congenital anomalies is 23.9 per 1000 births [1]. Ultrasonography

**Data availability statement:** All relevant data are within the manuscript.

**Funding:** The author(s) received no specific funding for this work.

**Competing interests:** The authors have declared that no competing interests exist.

during pregnancy is the key examination used to identify those anomalies. Fetal developmental anomalies arise from various causes, among which intrinsic causes (chromosomal anomalies, copy number variants or single nucleotide variants), maternal factors (such as age, diabetes, obesity, and stress), and environmental factors have been extensively reported. Maintaining TL is crucial for cell division during embryogenesis and organogenesis, but there is a lack of research examining the impact of TL variations on the occurrence of congenital anomalies in humans. [2,3].

Telomeres are nucleoprotein structures at the ends of chromosomes, consisting of non-coding repeated TTAGGG sequences and associated proteins that protect chromosome ends and preserve genomic stability [4]. In somatic cells, telomere length (TL) decreases with each cell cycle, leading to cellular senescence and aging. Conversely, germ cells, embryonic tissue cells, mechanism by which TL influences embryonic and fetal development is largely unknown. It is well established that fetuses are subjected to numerous maternal exposures, such as stress, tobacco, and socioeconomic status, which can impact their TL [5–8]. Derradji et al. [9] studied certain mechanisms involved in the occurrence of upper limb anomalies in mice and suggested a potential link between radiation exposure, telomere shortening dynamics, and the onset of development anomalies. Short telomeres were reported in four children with limb malformations, suggesting a possible link between TL and congenital malformations [2]. Two studies, conducted by the same team, have also revealed a link between decreased maternal TL and the occurrence of fetal spina bifida and ventricular septal defects in mice and humans [10,11]. In our previous study, we showed that telomere length (TL) was notably reduced in amniotic fluid (AF) when the fetus exhibited a developmental anomaly [12]. AF is known to contain multiple cell types and stem cells derived from the developing fetus [13,14]. Thus, human AF has been used in prenatal diagnosis for a number of years, and by screening this fluid and the cells contained within it, a variety of genetic and developmental disorders of the fetus can be diagnosed.

The objective of this study was to determine whether there are associations between maternal leukocyte TL and the risk of congenital malformations in fetuses, and whether there is a correlation between maternal and fetal TL in the context of abnormal development.

## Materials and methods

### Ethics statement

This retrospective, monocentric study used leftover samples from routine diagnostic tests. Written informed consent was obtained from patients for the use of these residual samples in research. All samples and data were anonymized. The study received approval from the local Institutional Review Board (IRB00013040, CPP SUD-EST VI Clermont-Ferrand, 2021/CE17).

### Sample collection

Amniotic fluid (AF) samples used for fetal TL assessment were collected by obstetricians between January 2022 and April 2024 following standard clinical procedures for

prenatal chromosomal diagnosis. The AF samples were obtained between 15.7 and 37.3 weeks of gestation, depending on the indication.

Exclusion criteria were infectious diseases, preeclampsia, diabetes, heavy smoking (> 10 cigarettes per day), high blood pressure, multifetal pregnancies and pregnancies with mechanical factors such as amniotic bands, anhydramnios or severe oligohydramnios.

To assess maternal TL, peripheral blood samples were obtained from pregnant women at the time of prenatal diagnosis, and also from controls.

### TL measurement using quantitative PCR

AF and blood samples were stored at −20°C until DNA extraction, performed with a Maxwell® 16 Instrument following the manufacturer's instructions (Promega, Charbonnières-les-Bains, France). DNA purity and concentration were assessed with a NanoDrop 1000 spectrophotometer (ThermoScientific, Wilmington, DE, USA). Quantitative PCR were performed using the same methods for all samples, as described in our previous study [12]. TL was measured by quantitative PCR in a LightCycler 480 System (Roche Diagnostics, Meylan, France) using SYBR Green I technology (SYBR Green Kit, Roche Diagnostics). For each sample, the results were expressed as the total length of telomeric DNA sequences, measured in kb per diploid genome.

### Statistical analysis

All statistical analyses were performed with GraphPad 10.2.3 software. Linear regression was used to assess the correlation between TL and maternal age. The power of the test was calculated based on Fisher's z-transformation. The normality of TL distribution across different subgroups was confirmed by the Kolmogorov-Smirnov test, and comparisons were performed using the t-test. A receiver operating characteristic (ROC) analysis was performed to assess the ability of maternal TL to discriminate between cases with and without congenital anomalies. The optimal TL cut-off value was determined using the Youden index. A p-value of less than 0.05 was considered statistically significant.

## Results

### Study cohort

A total of 75 evolutive pregnancies with abnormal fetal development were included in the study (Fig 1). Fetal congenital anomalies (CA) include cleft lip and palate (n = 1), limb malformations (n = 5), cardiac malformations (n = 27), digestive malformations (n = 3), spina bifida (n = 4), diaphragmatic hernia (n = 6), urinary tract anomaly (n = 1) brain malformations (n = 9) and multiple malformations (n = 19). All fetuses had normal karyotype and chromosomal microarray (Agilent 180K, threshold 400 kb). In addition, exome sequencing was performed on 23 fetuses, and the results were normal. In parallel, we evaluated maternal TL in DNA from peripheral blood leukocytes obtained at the time of a prenatal diagnosis for the 75 pregnancies affected with CA. Pregnant women with fetuses presumed healthy (n = 50), and non-pregnant women who had previously one or more children without any congenital malformation (n = 50) formed two control groups for maternal leukocyte TL (Fig 1). Control AF samples (n = 49) were obtained from a separate cohort of pregnant women with normal fetal ultrasounds who were undergoing prenatal diagnosis owing to advanced maternal age or a high biological risk of trisomy 21 (Fig 1).

### Fetal telomere length and developmental anomalies

Compared to control fetuses, TL in AF samples was significantly (P < 0.0001) shorter in fetuses with isolated and multiple malformations (Fig 2A).

**Maternal blood sample**

**Control 1: Pregnant women with no abnormal fetal development (N= 50)**
Maternal age: 37.2±5.7   Gestational age: 16.0±4.3

**Control 2: Non-pregnant women with one or more children without developmental disorders (N= 50)**
Maternal age: 31.9±5.9

**Pregnant woman with abnormal fetal development (N=75)**
Maternal age: 31.8±5.6  Gestational age : 26.1±4.6

**Fetal sample (Amniotic fluid)**

**Control: fetal normal ultrasound (N= 49)**
Maternal age: 34.7±6.0  Gestational age : 20.4±5.3

**Fetal abnormal ultrasound (N=75)**

▪ **Isolated malformation (N=56)**
Maternal age: 32.4±5.9   Gestational age: 25.6±4.8

▪ **Multiple malformation (N= 19)**
Maternal age: 32.2±5.3   Gestational age: 25.3±4.7

**Fig 1. Study population.** A total of 175 maternal blood samples and 124 amniotic fluid samples were analyzed.

Comparison of the TL for each type of malformation with that in control patients showed a significant reduction in TL for all types of developmental anomaly. The only exception was a fetus with a cleft lip and palate, in which no significant difference in TL was observed (Fig 2B). Fetuses with multiple congenital anomalies had significantly shorter telomeres than those with isolated malformations (P < 0.001).

### Age-adjusted maternal leukocyte telomere length

As expected, a significant decrease in telomere length with age was observed in the control (Fig 3A, $r = -0.58$, power of 0.99) and pathological (Fig 3B, $r = -0.32$, power of 0.80) groups. This relationship was also studied in our laboratory in an additional control cohort of female subjects aged 0.1 to 94 years showing a strong inverse correlation between the age and TL (Fig 3C, $r = -0.74$, power of 0.99). The linear regression equation derived from this cohort allowed us to estimate the expected TL at a given age in our study population. To adjust maternal TL for age, we calculated the ratio between observed and expected TL. The age-adjusted TL values in our study populations are shown in Fig 3D. Age-adjusted TL did not differ significantly between pregnant and non-pregnant control women (P = 0.24). Notably, both groups had median ratios of 1.0, indicating that their TLs were consistent with their chronological age. In contrast, mothers carrying fetuses with isolated or multiple congenital anomalies had significantly lower median ratios of 0.78 and 0.75, respectively (P < 0.0001), indicating that their TLs were shorter than expected for their age.

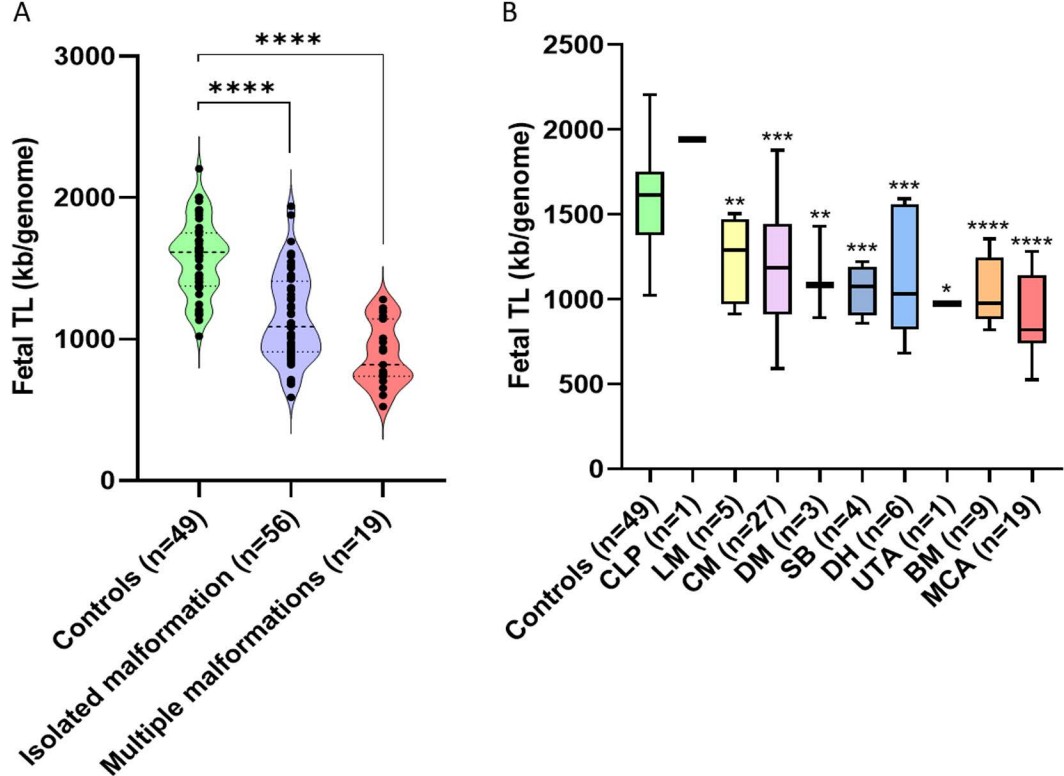

**Fig 2. Telomere length (TL) in amniotic fluid (AF) samples.** A. TL in AF samples from control fetuses and pathological fetuses with isolated or multiple malformations. B. TL in AF according to the type of developmental anomaly (CLP: cleft lip and palate, LM: limb malformation, CM: cardiac malformation, DM: digestive malformation, SB: spina bifida, DH: diaphragmatic hernia, UTA: urinary tract anomaly, BM: brain malformation, MCA: multiple congenital anomalies). P-values: * < 0.05, ** < 0.01, *** < 0.001, **** < 0.0001.

To assess the ability of age-adjusted maternal TL to discriminate between cases with and without congenital anomalies, we performed a ROC analysis on control (n = 100) and affected (n = 75) pregnancies (Fig 3E). The area under the ROC curve (AUC) was 0.86, indicating a high overall discriminative ability of maternal TL for identifying congenital anomalies (P < 0.0001). The optimal TL cut-off value of 0.874 identified cases of congenital anomalies with 92% specificity and 73% sensitivity.

## Correlation between maternal and fetal TL

To strengthen this result, we focused on fetuses with developmental anomalies and the shortest telomeres. We set a threshold at 1028 kb, corresponding to the median TL in AF. Notably, none of the control fetuses had a TL below this threshold. We defined two groups among fetuses with developmental anomalies, one with fetal TL ≥ 1028 kb/genome and one with fetal TL < 1028 kb/genome (Fig 4A). We then analyzed maternal age-adjusted TL in each group, which confirmed that maternal TL was significantly shorter in the group with very short fetal telomeres (P < 0.01) (Fig 4B). These results suggest a potential heritability of TL, which would play a significant role in the occurrence of fetal developmental abnormalities.

We explored the relationship between fetal TL (AF) and maternal TL (leukocytes) in cases of fetal developmental anomalies. The correlation analysis revealed significant determination coefficients across clinical categories: isolated anomalies $R^2$ = 0.15 (P = 0.0035, power of 0.83) and multiple anomalies $R^2$ = 0.38 (P = 0.005, power of 0.82) (Fig 4C).

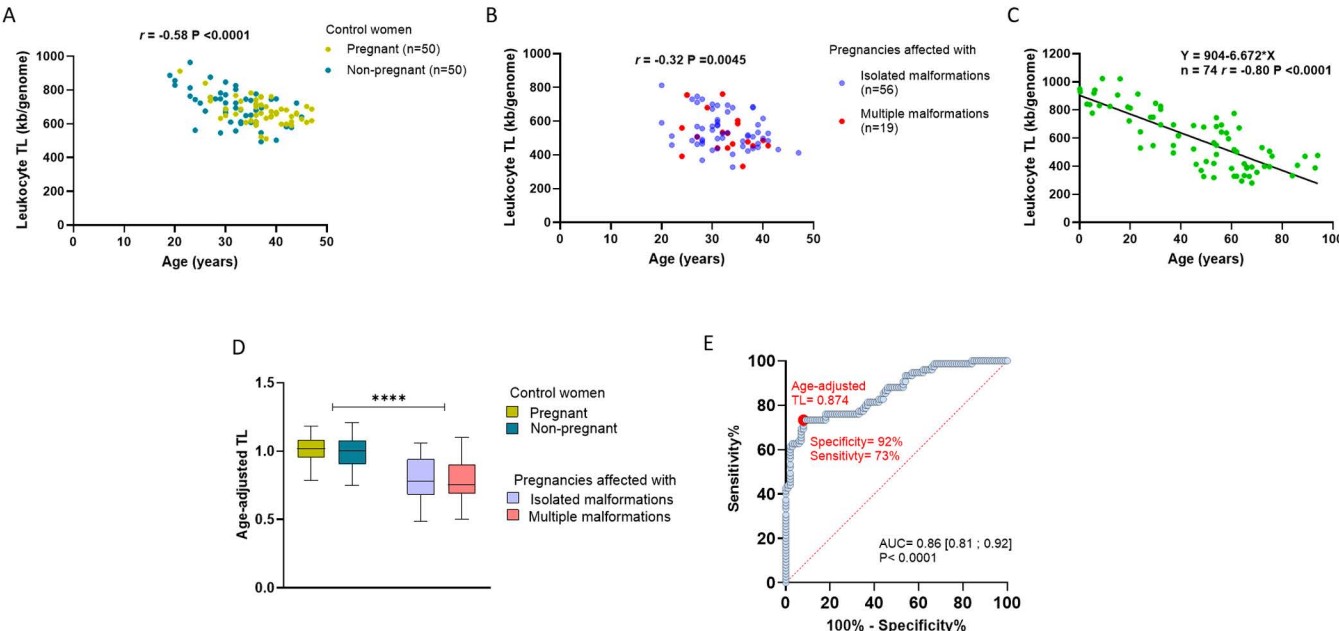

**Fig 3. Telomere length (TL) in maternal blood samples.** A. Age-dependent decrease in the leukocyte TL in the control groups of the study, comparing pregnant and non-pregnant women. B. Age-dependent decrease in the maternal leucocyte TL in pregnancies affected with developmental anomalies. C. Leukocyte TL as a function of age in healthy female donors aged 0.1 to 94 years (n = 74) from the control cohort of our cytogenetic department. The calibration curve represents the expected TL values across age and was used as a reference to calculate age-adjusted TL. D. Age-adjusted TL in the study cohorts. TL was similar between pregnant and non-pregnant controls (median = 1.0; P = 0.24). Mothers of fetuses with isolated or multiple defects showed significantly shorter TLs (medians = 0.78 and 0.75; P < 0.0001). E. A receiver operating characteristic (ROC) analysis of age-adjusted maternal TL for identifying congenital anomalies. An area under the curve (AUC) of 0.86 shows strong predictive accuracy. The optimal cut-off value 0.874, which showed a specificity of 92% and sensitivity of 73%.

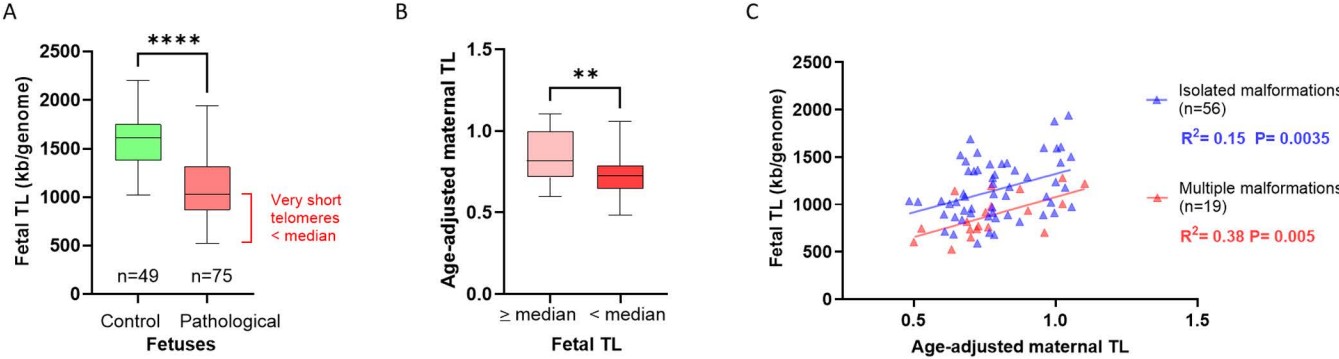

**Fig 4. Correlations between maternal and fetal TL.** A. Determination of a 'very low' fetal telomere length (TL) threshold based on the median TL in fetuses with developmental anomalies (1028 kb/genome). B. Age-adjusted maternal telomere length (TL) in the two previously defined groups: fetuses with telomeres ≧ 1028 kb/genome, and fetuses with telomeres < 1028 kb/genome. C. Correlations between maternal and fetal TL according to the type of developmental anomaly.

The coefficient of determination reflects the proportion of variance in fetal TL that is statistically accounted for by maternal TL. $R^2$-values indicate that maternal TL explains 15% of the variability in fetal TL in cases with isolated anomalies ($R^2$ = 0.15), and 38% in cases involving multiple congenital anomalies ($R^2$ = 0.38). Thus, maternal TL appears to be a stronger predictor of fetal TL in the context of multiple congenital malformations.

## Discussion

Despite advances in genetic testing technologies such as chromosomal microarray analysis and exome sequencing, a definitive diagnosis is still not achieved in over 40% of fetal structural anomaly cases detected by prenatal ultrasound. Exome sequencing can increase the diagnostic yield in fetuses with structural anomalies (from 5% to 57%, depending on the study) [15]. However, there are limitations to its use in the prenatal setting, including turnaround time, the reporting and interpretation of secondary and inconclusive findings, cost, and the potential for incomplete phenotypic information, which may hinder genotype–phenotype correlation [18].

A number of studies in humans and mice have indicated a correlation between a short TL and developmental abnormalities [2,3,9,16]. These studies predominantly focused on limb and central nervous system malformations.

Our results confirm that TL in AF was significantly lower in fetuses with either isolated or multiple malformations. With the exception of one fetus with a cleft lip and palate, a minor malformation that occurs late in embryonic development, TL was significantly lower in each type of malformation: heart defect, spina bifida, central nervous system anomaly, limb malformations, and diaphragmatic hernia.

Our study also shows that TL in mothers carrying fetuses with developmental anomalies is significantly lower than in the control group. This shortening of maternal TL could reflect an underlying biological state predisposing to pregnancy complications. Aoulad Fares et al. [10] suggest that preconception exposure to environmental and lifestyle risk factors accelerates a woman's ageing process, which can be measured by TL, and increases her underlying risk of having offspring with neural tube defects. Shorter periconception maternal TL could also contribute to the development of ventricular septal defects [11]. In the context of pregnancy, maternal TL appears to reflect the general susceptibility to developmental anomalies and could serve as a potential biomarker for identifying pregnancies at risk of developing congenital malformations. Pregnant women with short telomeres may benefit from closer pregnancy monitoring and preventive measures such as smoking cessation, folic acid supplementation.

Our data also indicated that there was no significant difference in leukocyte TL between the control groups of pregnant and non-pregnant women. This suggests pregnancy itself does not significantly affect maternal TL, which is consistent with the findings of previous studies showing that pregnancy does not necessarily lead to major changes in TL [17–19].

Finally, our results demonstrated a significant correlation between maternal and fetal TL in cases of developmental anomalies, particularly when multiple malformations were present. Other studies have also suggested the heritability of maternal TL [20–22]. If heritability is confirmed then periconceptional maternal TL could predispose not only to spontaneous miscarriage or preterm birth, as observed in the study of Han, et al. [23], but also influence the development and health of the child (Fig 5).

It should be emphasized it is likely that maternal TL is only partially responsible for the occurrence of developmental anomalies, since other environmental, genetic, and epigenetic factors contribute to telomere shortening [5,6,8] (Fig 5). Our study therefore has certain limitations, including the sample size and the need to consider other potential factors such as environmental exposures. Additionally, it would have been interesting to study paternal TL as well, as in the study of Chen et al. [20], to analyze further possible influences on the appearance of fetal developmental anomalies.

In conclusion, this study confirms the association between shortened maternal and fetal telomeres and occurrence of malformations, raising the possibility that maternal telomere length during the periconceptional period could be used as a predictive biomarker for the occurrence of developmental anomalies.

However, it is crucial to note that correlation does not imply causation. Additional studies, such as the search for genomic variants in genes involved in telomere maintenance, are needed to elucidate the mechanisms involved in maternal and fetal telomere shortening and in the occurrence of developmental anomalies. The measurement of TL could thus be used as an additional tool in the diagnostic strategy for fetal malformations, by orienting the analysis of exome or genome sequencing data towards the search for variants in telomere maintenance genes.

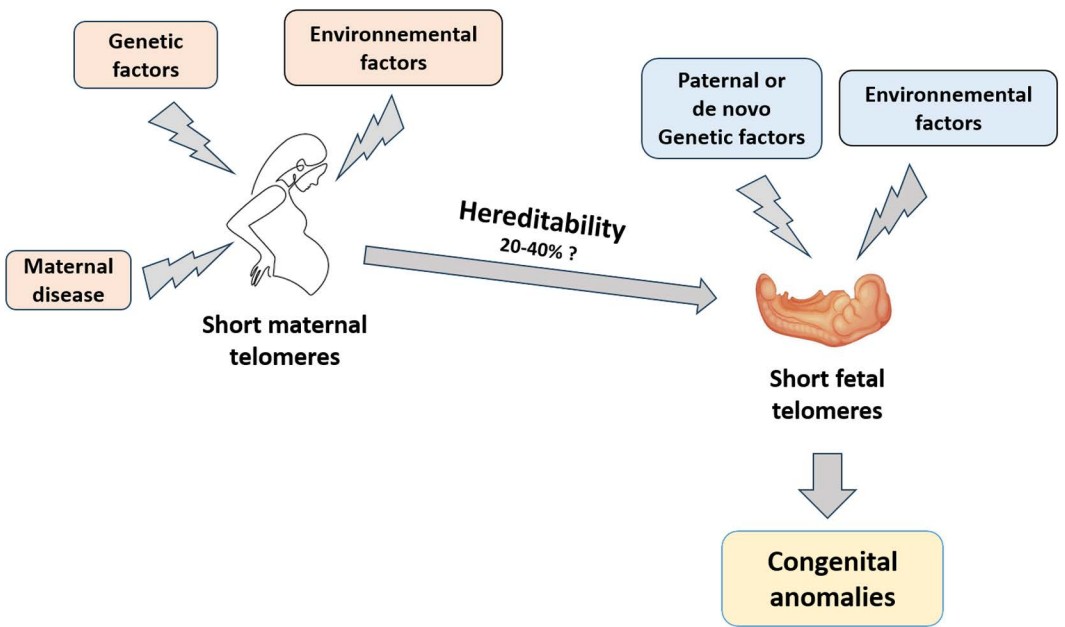

**Fig 5. Schematic diagram illustrating the potential role of maternal and fetal telomere length in congenital anomalies and intrauterine growth retardation, possibly modulated by various environmental and hereditary factors.**

## Acknowledgments

The authors thank the patients for their consent and Delphine Voisin for expert technical assistance.

## Author contributions

**Conceptualization:** Carole Goumy.

**Data curation:** Carole Goumy, Océane Coudrieu, Andrei Tchirkov.

**Formal analysis:** Andrei Tchirkov.

**Investigation:** Océane Coudrieu, Denis Gallot, Amélie Delabaere, Eleonore Eymard-Pierre.

**Methodology:** Carole Goumy.

**Supervision:** Andrei Tchirkov.

**Writing – original draft:** Carole Goumy.

**Writing – review & editing:** Zangbéwendé Guy Ouedraogo, Lauren Veronese, Andrei Tchirkov.

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
