## [Decision Letter · Decision Letter 0]

PONE-D-25-04698
Association between shortened maternal and fetal telomere length and abnormal fetal development
PLOS ONE

Dear Dr. GOUMY,

Thank you for submitting your manuscript to PLOS ONE. After careful consideration, we feel that it has merit but does not fully meet PLOS ONE’s publication criteria as it currently stands. Therefore, we invite you to submit a revised version of the manuscript that addresses the points raised during the review process, detailed below.

**Would you consider the proposed, major revision of your manuscript, **please submit your revised manuscript by Jun 07 2025 11:59PM. If you will need more time than this to complete your revisions, please reply to this message or contact the journal office at plosone@plos.org. Please include the following items when submitting your revised manuscript:

We look forward to receiving your revised manuscript.

Kind regards,

Umberto Simeoni

Academic Editor

PLOS ONE

**Journal Requirements:**

1. When submitting your revision, we need you to address these additional requirements.
 
Please ensure that your manuscript meets PLOS ONE's style requirements, including those for file naming. The PLOS ONE style templates can be found at 
https://journals.plos.org/plosone/s/file?id=wjVg/PLOSOne_formatting_sample_main_body.pdf and 
https://journals.plos.org/plosone/s/file?id=ba62/PLOSOne_formatting_sample_title_authors_affiliations.pdf
 
2. Please upload a new copy of Figures 1 to 5 as the detail is not clear. Please follow the link for more information: https://blogs.plos.org/plos/2019/06/looking-good-tips-for-creating-your-plos-figures-graphics/" https://blogs.plos.org/plos/2019/06/looking-good-tips-for-creating-your-plos-figures-graphics/

**Additional Editor Comments :**

Despite the originality of the topic, major issues affect the manuscript, relating mainly in a somehow confusing study design, statistical analysis and interpretation.

:
1. The current study design includes both a population of fetuses with congenital malformations, and in a different, smaller group IUGR fetuses, with a restrictive definition (Extremely low percentile definition, exclusion of preeclampsia,...). Such conditions being different, heterogeneous, and only partially overlapping in terms of mechanisms , the robustness of any conclusion drawn from the study results is questionable. Restricting the study hypothesis to pregnancies with fetal congenital abnomalies for example would be preferable, while IUGR might be the object of a separate study.
2. The remark n°1 above is even more important as the hypothesis made by the authors seems to focus on telomere length  (TL) shortening as a causal, eventually genetic and heritable factor, which is possible, but not considering in the analysis the hypothesis that TL might be a consequence of acquired, environmental mechanisms, affecting both the other and the fetuses.
3. The manuscript defines TL as a potential biomarker for congenital abnomalies and IUGR together, grouped as developmental disorders. However, the statistical analysis does not include the criteria defining a biomarker, such as specificity, sensitivity, positive and negative predictive values and likelyhood ratios, or ROC curves. This is a major drawback which makes the conclusions in the manuscript not supported by the study design and the results.
Minor remarks:
1. The cellular origin of amniotic fluid TL determination is not explicited.
2. The legends to the tables/figures should be located in a separate section of the manuscript
3. The significance of the abbreviation "MCA", which appears in the text, needs to be searched for in a figure legend.

Reviewers' comments:

Reviewer's Responses to Questions

**Comments to the Author**

1. Is the manuscript technically sound, and do the data support the conclusions?

Reviewer #1: Yes

Reviewer #2: Yes

2. Has the statistical analysis been performed appropriately and rigorously? 

Reviewer #1: Yes

Reviewer #2: Yes

3. Have the authors made all data underlying the findings in their manuscript fully available?

Reviewer #1: Yes

Reviewer #2: Yes

4. Is the manuscript presented in an intelligible fashion and written in standard English?

Reviewer #1: Yes

Reviewer #2: Yes

5. Review Comments to the Author

**Reviewer #1: **Your paper is a meaningful work, with a good technical methodology and justification, but I have not understood some points or have some notes.

First, I think it is necessary to pinpoint that the pregnancy is evolutive, and has many different exposures because of it, mainly for IUGR. So, why did you choose some comparisons to be made between gestational ages different, such as amniotic fluid (20 GW, for controls, vs. 25 GW, for cases), or maternal blood (16 GW, for controls, vs. 26 GW, for cases)? At these different frames you could lost of paramount events on placentation, such as uterine vessels transformation.

Second, IUGR use to have a component of placental insufficiency, so it may reflect your measurements being more disperse on this category, but how can you screen it with TL only? I think you should discuss the limitations of TL test or study for a cut-off using gestational age because of factors, such as infections, could alter the length (doi: 10.1186/s12967-024-05879-0). Maybe, also, could it be adjunct to NIPT test on future?

Third, on page 6, lines 24-25, I missed a reference. On page 8, lines 1-4, the extrapolation of data was not fully clear to me. Then, I think you should explain it better.

**Reviewer #2:** The article is interesting, but some adjustments and more detailed elaboration are needed in both the introduction and the discussion of the results.

1- Introduction:

1.1 The phrase "chromosomal or molecular variations" is not sufficiently precise, I recommend using more specific terminology, such as copy number variants (CNVs) for structural alterations and single nucleotide variants (SNVs)

1.2 It would be useful if the authors provided more detailed background on previous studies that investigate the association between telomere length and fetal development.

2- Statistycal analysis

The authors should clarify whether the sample size was sufficient to detect a significant correlation in the regression analysis. Including a justification or a power calculation would strengthen the statistical validity of the results.

3- Results:

3.1- The paper uses the term IUGR, which has been largely replaced in current literature by Fetal Growth Restriction (FGR). I recommend updating the terminology to FGR, in line with recent guidelines in obstetrics and fetal medicine.

3.2- The phrase "All fetuses had normal karyotype and array CGH results (Agilent 180K, threshold 400 kb)" could be interpreted as ambiguous. It is unclear whether the array CGH results were also normal. I recommend clarifying this sentence.

The authors should also point out whether some of these cases underwent NGS analysis (exome sequencing).

3.3: The authors should avoid using the expression in bold: “..prenatal diagnosis for the 108 pregnancies affected with CA” since only 75 fetuses had congenital anomalies, while the other 33 had severe FGR.

3.4 : In the sentence "Fetuses with MCA had significantly shorter telomeres than those with isolated 13 malformations (P<0.001)," the abbreviation MCA has not been defined earlier in the text. I suggest the authors to use abbreviations only when strictly necessary.

3.5. Please clarify the abbreviation RTL and LT ( or the correct term is TL?) since it was used only once: “We then analyzed maternal age-adjusted TL in each group, which confirmed that maternal RTL was significantly shorter…” …These results suggest a potential heritability of LT…

4- Discussion

4.1 – The authors should consider discussing the potential implications and limitations of not performing exome sequencing in cases of multisystem and brain anomalies since it is one of the indications of ES in prenatal diagnostic (many references about this topic); and also to discuss better this topic.

4.2 The sentence "Their findings suggest that TL in prenatal samples could be a useful indicator for assessing the risk of developmental anomalies in fetuses" might be somewhat overestimating the role of TL, given that morphological screening in the first trimester already detects around 40% of malformations. I would suggest rephrasing this to reflect that TL could be considered as an additional tool in assessing risk, particularly in maternal samples, rather than a primary indicator.

Figures:

The figures have very blurred lettering. I suggest using a version with better visualization.

6. PLOS authors have the option to publish the peer review history of their article (what does this mean?). If published, this will include your full peer review and any attached files.

Reviewer #1: No

Reviewer #2: No

---

## [Author Response · Author response to Decision Letter 1]

5 Jun 2025

Additional Editor Comments:

Despite the originality of the topic, major issues affect the manuscript, relating mainly in a somehow confusing study design, statistical analysis and interpretation.

1. The current study design includes both a population of fetuses with congenital malformations, and in a different, smaller group IUGR fetuses, with a restrictive definition (Extremely low percentile definition, exclusion of preeclampsia,...). Such conditions being different, heterogeneous, and only partially overlapping in terms of mechanisms , the robustness of any conclusion drawn from the study results is questionable. Restricting the study hypothesis to pregnancies with fetal congenital abnomalies for example would be preferable, while IUGR might be the object of a separate study.

As you suggested, we have excluded the population of fetuses with growth retardation.

2. The remark n°1 above is even more important as the hypothesis made by the authors seems to focus on telomere length (TL) shortening as a causal, eventually genetic and heritable factor, which is possible, but not considering in the analysis the hypothesis that TL might be a consequence of acquired, environmental mechanisms, affecting both the other and the fetuses.

Short telomere may be the result of acquired environmental factors, and is likely to be multifactorial. Congenital short telomeres may also make the fetus more sensitive to environmental factors.

3. The manuscript defines TL as a potential biomarker for congenital abnomalies and IUGR together, grouped as developmental disorders. However, the statistical analysis does not include the criteria defining a biomarker, such as specificity, sensitivity, positive and negative predictive values and likelyhood ratios, or ROC curves. This is a major drawback which makes the conclusions in the manuscript not supported by the study design and the results.

We have performed a receiver operating characteristic (ROC) analysis to assess the use of maternal TL as a reliable discriminating factor regarding the existence of congenital anomalies. In the abstract and the results section, we have underlined the 92% specificity and 73% sensitivity of the ROC analysis at the optimal cut-off value.

Minor remarks:

1. The cellular origin of amniotic fluid TL determination is not explicited.

We have clarified the origin of the amniotic fluid cells in the introduction section (highlighted manuscript: page 4, lines 2-5)

2. The legends to the tables/figures should be located in a separate section of the manuscript

The figure legends are now in a separate section at the end of the manuscript.

3. The significance of the abbreviation "MCA", which appears in the text, needs to be searched for in a figure legend.

We have deleted the abbreviation “MCA” in the text and replaced it with “multiple congenital anomalies”.

Reviewer 1

Reviewer #1: Your paper is a meaningful work, with a good technical methodology and justification, but I have not understood some points or have some notes.

1/ First, I think it is necessary to pinpoint that the pregnancy is evolutive, and has many different exposures because of it, mainly for IUGR.

We have specified in the abstract and in the results section that the pregnancies were evolutive (page 6, line 1).

2/ So, why did you choose some comparisons to be made between gestational ages different, such as amniotic fluid (20 GW, for controls, vs. 25 GW, for cases), or maternal blood (16 GW, for controls, vs. 26 GW, for cases)? At these different frames you could lost of paramount events on placentation, such as uterine vessels transformation. Second, IUGR use to have a component of placental insufficiency, so it may reflect your measurements being more disperse on this category, but how can you screen it with TL only? I think you should discuss the limitations of TL test or study for a cut-off using gestational age because of factors, such as infections, could alter the length (doi: 10.1186/s12967-024-05879-0). Maybe, also, could it be adjunct to NIPT test on future?

We did not perform comparisons between different gestational ages. In our previous study (Goumy et al., 2022), we demonstrated that gestational age had no impact on TL in chorionic villi and amniotic fluid, in either normal pregnancies or in those with developmental anomalies (including IUGR). Therefore, we did not select samples based on gestational age.

As suggested by the Editor’s recommendations, we have excluded the population of fetuses with growth restriction, particularly for the reasons you mentioned.

3/ Third, on page 6, lines 24-25, I missed a reference: “This relationship was previously studied in our laboratory in a control cohort of female subjects aged 0.1 to 94 years (Figure 3C).”

These results have not been published. This relates to a control cohort of female subjects that we used in our laboratory to interpret the results of TL measurement.

We have revised the sentence and included an added one to clarify this point: ”This relationship was also studied in our laboratory in an additional control cohort of female subjects aged 0.1 to 94 years showing a strong inverse correlation between the age and TL (Figure 3C, r= -0.74, power of 1). The linear regression equation derived from this cohort allowed us to estimate the expected TL at a given age in our study population.”

4/ On page 8, lines 1-4, the extrapolation of data was not fully clear to me. Then, I think you should explain it better. “The correlation analysis showed significant determination coefficients across clinical categories: isolated anomalies R2=0.15 P= 0.0063, multiple anomalies R2=0.38, P=0.005, and IUGR R2=0.22, P=0.0035 (Figure 4C). These results indicate that 15% to 38% of the variance in fetal TL is attributable to maternal TL, particularly in fetuses with multiple congenital malformations.”

We have edited the sentences, specified the power and explained these results: “The correlation analysis revealed significant determination coefficients across clinical categories: isolated anomalies R2= 0.15 (P= 0.0035, power of 0.83) and multiple anomalies R2= 0.38 (P= 0.005, power of 0.82). (Figure 4C). The coefficient of determination reflects the proportion of variance in fetal TL that is statistically accounted for by maternal TL. R2-values indicate that maternal TL explains 15% of the variability in fetal TL in cases with isolated anomalies (R2= 0.15), and 38% in cases involving multiple congenital anomalies (R2= 0.38). Thus, maternal TL appears to be a stronger predictor of fetal TL in the context of multiple congenital malformations.” (page 8, lines 7-14)

Reviewer 2

Reviewer #2: The article is interesting, but some adjustments and more detailed elaboration are needed in both the introduction and the discussion of the results.

1- Introduction:

1.1 The phrase "chromosomal or molecular variations" is not sufficiently precise, I recommend using more specific terminology, such as copy number variants (CNVs) for structural alterations and single nucleotide variants (SNVs)

We have used more specific terminology to describe chromosomal and molecular abnormalities: “chromosomal anomalies, copy number variants or single nucleotide variants” (Page 3, lines: 6-7)

1.2 It would be useful if the authors provided more detailed background on previous studies that investigate the association between telomere length and fetal development.

We have added data and bibliographical references concerning studies showing an association between fetal development and telomere length in the introduction section. (Page 3, lines: 21-25 and page 4 lines 1-2)

2- Statistycal analysis

The authors should clarify whether the sample size was sufficient to detect a significant correlation in the regression analysis. Including a justification or a power calculation would strengthen the statistical validity of the results.

We calculated the statistical power to strengthen the validity of our results: “The power of the test was calculated based on Fisher’s z-transformation” (page 5, lines 15–16). This analysis indicated that our sample sizes allowed us to achieve power values ranging from 0.8 to 0.99.

3- Results:

3.1- The paper uses the term IUGR, which has been largely replaced in current literature by Fetal Growth Restriction (FGR). I recommend updating the terminology to FGR, in line with recent guidelines in obstetrics and fetal medicine.

We have excluded the 33 pregnancies with fetal growth restriction in accordance with the Editor's recommendations.

3.2- The phrase "All fetuses had normal karyotype and array CGH results (Agilent 180K, threshold 400 kb)" could be interpreted as ambiguous. It is unclear whether the array CGH results were also normal. I recommend clarifying this sentence.

The authors should also point out whether some of these cases underwent NGS analysis (exome sequencing).

Array CGH results were also normal.

Exome sequencing was performed on 23 fetuses. It is conducted as a second-line investigation when the microarray results are normal, provided that the parents wish to continue the pregnancy and consent to further testing.

We have now included the exome sequencing results in the text: “All fetuses had normal karyotype and chromosomal microarray (Agilent 180K, threshold 400 kb). In addition, exome sequencing was performed on 23 fetuses, and the results were normal.” (Page 6, lines: 6-9).

3.3: The authors should avoid using the expression in bold: “..prenatal diagnosis for the 108 pregnancies affected with CA” since only 75 fetuses had congenital anomalies, while the other 33 had severe FGR.

In accordance with the Editor’s recommendations, we have excluded the 33 pregnancies with fetal growth restriction.

3.4 : In the sentence "Fetuses with MCA had significantly shorter telomeres than those with isolated 13 malformations (P<0.001)," the abbreviation MCA has not been defined earlier in the text. I suggest the authors to use abbreviations only when strictly necessary.

We have deleted the abbreviation “MCA” in the text and replaced it with “multiple congenital anomalies”. (Page 6, lines: 23)

3.5. Please clarify the abbreviation RTL and LT (or the correct term is TL?) since it was used only once: “We then analyzed maternal age-adjusted TL in each group, which confirmed that maternal RTL was significantly shorter…” …These results suggest a potential heritability of LT…

We have replaced the abbreviations “RTL” and “LT” with “TL”. (Page 8, lines: 3-4)

4- Discussion

4.1 – The authors should consider discussing the potential implications and limitations of not performing exome sequencing in cases of multisystem and brain anomalies since it is one of the indications of ES in prenatal diagnostic (many references about this topic); and also to discuss better this topic.

We performed exome analyses on 23 patients (see addition in study cohort section: page 6, lines 7-8). We have revised the beginning of the Discussion section and added a sentence addressing the potential implications and limitations of not performing exome sequencing: “Exome sequencing can increase the diagnostic yield in fetuses with structural anomalies (from 5% to 57%, depending on the study). However, there are limitations to its use in the prenatal setting, including turnaround time, the reporting and interpretation of secondary and inconclusive findings, cost, and the potential for incomplete phenotypic information, which may hinder genotype–phenotype correlation [18].” (Page 8, lines 19–23)

4.2 The sentence "Their findings suggest that TL in prenatal samples could be a useful indicator for assessing the risk of developmental anomalies in fetuses" might be somewhat overestimating the role of TL, given that morphological screening in the first trimester already detects around 40% of malformations. I would suggest rephrasing this to reflect that TL could be considered as an additional tool in assessing risk, particularly in maternal samples, rather than a primary indicator.

We have deleted this sentence: “Their findings suggest that TL in prenatal samples could be a useful indicator for assessing the risk of developmental anomalies in fetuses.”

We have replaced it with the following sentence in the conclusion: “The measurement of TL could thus be used as an additional tool in the diagnostic strategy for fetal malformations, by orienting the analysis of exome or genome sequencing data towards the search for variants in telomere maintenance genes.” (Page 10, lines: 19-22)

Figures:

The figures have very blurred lettering. I suggest using a version with better visualization.

We have improved the quality of the figures.

---

## [Decision Letter · Decision Letter 1]

Association between shortened maternal and fetal telomere length and abnormal fetal development

PONE-D-25-04698R1

Dear Dr. GOUMY,

We’re pleased to inform you that your manuscript has been judged scientifically suitable for publication and will be formally accepted for publication once it meets all outstanding technical requirements.

Kind regards,

Umberto Simeoni

Academic Editor

PLOS ONE

Additional Editor Comments (optional):

Reviewers' comments:

Reviewer's Responses to Questions

**Comments to the Author**

1. If the authors have adequately addressed your comments raised in a previous round of review and you feel that this manuscript is now acceptable for publication, you may indicate that here to bypass the “Comments to the Author” section, enter your conflict of interest statement in the “Confidential to Editor” section, and submit your "Accept" recommendation.

Reviewer #1: All comments have been addressed

Reviewer #2: All comments have been addressed

2. Is the manuscript technically sound, and do the data support the conclusions?

Reviewer #1: Yes

Reviewer #2: Yes

3. Has the statistical analysis been performed appropriately and rigorously? 

Reviewer #1: Yes

Reviewer #2: Yes

4. Have the authors made all data underlying the findings in their manuscript fully available?

Reviewer #1: Yes

Reviewer #2: Yes

5. Is the manuscript presented in an intelligible fashion and written in standard English?

Reviewer #1: Yes

Reviewer #2: Yes

6. Review Comments to the Author

Reviewer #1: Your article is very interesting and you have adressed every comment made about improvements. The text is more clear, and I think that it is going to prone to indicate telomere length as a pre-conceptual and pre-natal tool, as you discussed.

I encourage you to publish a comparison with fetal growth restriction and small-for-gestacional-age, look for properly avoid selection biases.

Reviewer #2: Both the editor's and both reviewers' reviews improved the article, and the authors made appropriate changes.

7. PLOS authors have the option to publish the peer review history of their article (what does this mean?). If published, this will include your full peer review and any attached files.

Reviewer #1: No

Reviewer #2: No

---

## [Editor Report · Acceptance letter]

PONE-D-25-04698R1

PLOS ONE

Dear Dr. Goumy,

I'm pleased to inform you that your manuscript has been deemed suitable for publication in PLOS ONE. Congratulations! Your manuscript is now being handed over to our production team.

Kind regards,

on behalf of

Prof. Umberto Simeoni

Academic Editor

PLOS ONE